# Klein Tunneling in *β*_12_ Borophene

**DOI:** 10.3390/nano14090790

**Published:** 2024-05-01

**Authors:** Jinhao Lai, Lekang Wang, Fu Li, Hongbin Zhang, Qingtian Zhang

**Affiliations:** 1School of Materials and Energy, Guangdong University of Technology, Guangzhou 510006, China; 2112102173@mail2.gdut.edu.cn (J.L.); 2112202161@mail2.gdut.edu.cn (L.W.); 2Institute of Materials Science, Technology University of Darmstadt, 64287 Darmstadt, Germany; fu.li@tu-darmstadt.de (F.L.); hzhang@tmm.tu-darmstadt.de (H.Z.); 3Guangdong Provincial Key Laboratory of Information Photonics Technology, Guangdong University of Technology, Guangzhou 510006, China

**Keywords:** Klein tunneling, β_12_ borophene, Dirac fermions

## Abstract

Motivated by the recent observation of Klein tunneling in 8-Pmmn borophene, we delve into the phenomenon in β_12_ borophene by employing tight-binding approximation theory to establish a theoretical mode. The tight-binding model is a semi-empirical method for establishing the Hamiltonian based on atomic orbitals. A single cell of β_12_ borophene contains five atoms and multiple central bonds, so it creates the complexity of the tight-binding model Hamiltonian of β_12_ borophene. We investigate transmission across one potential barrier and two potential barriers by changing the width and height of barriers and the distance between two potential barriers. Regardless of the change in the barrier heights and widths, we find the interface to be perfectly transparent for normal incidence. For other angles of incidence, perfect transmission at certain angles can also be observed. Furthermore, perfect and all-angle transmission across a potential barrier takes place when the incident energy approaches the Dirac point. This is analogous to the “super”, all-angle transmission reported for the dice lattice for Klein tunneling across a potential barrier. These findings highlight the significance of our theoretical model in understanding the complex dynamics of Klein tunneling in borophene structures.

## 1. Introduction

Two-dimensional (2D) materials have attracted intense research interests owning to their unique physicochemical properties and potential applications in electronics [1,2], sensors [3,4], energy storage [5,6], optoelectronics [7,8], biomedicine [9,10], etc. The intense studies on 2D materials have begun with graphene [11] and extended to other group IV monolayer materials, such as germanene [12,13], silicene [14,15], and stanene [16,17]. Furthermore, a large number of other elemental 2D materials including phosphorene [18,19], arsenene [20,21], and tellurium [22] have been successfully produced. In addition, other new nanostructures and complex structures have also been reported, such as indium oxide in two-dimensional structures [23] and silicon cages in metal encapsulation [24]. Until now, the search for 2D materials has been extended to a wide range, and many novel 2D materials, such as the transition metal dichalcogenides [25], noble metal dichalcogenides [26], MXenes [27], and hexanol boron nitride [28], have been synthesized.

Recently, borophene, a monolayer of boron atoms arranged in a honeycomb-like lattice, has attracted a lot of attention [29,30]. Borophene was first theoretically predicted in the mid-1990s [31]; however, the realization of monoatomic-layer borophene was reported in 2005 to 2016 [32,33,34,35,36,37]. A large variety of monolayer boron structures, such as the α sheet, β sheet, and χ sheets have been proposed theoretically [33,34,38,39,40,41]. The α, β, and χ phases of borophene have been experimentally realized on Ag(111) under ultrahigh-vacuum conditions [42,43,44,45,46]. It was predicted in the previous experimental studies [36,37,44] that the most stable allotrope of borophene is *β*_12_ borophene, because *β*_12_ borophene has a flat structure that weakly interacts with the Ag(111) substrate. Thanks to the inherent stability of β_12_ borophene’s structure, its distinctive electronic properties and electron transport characteristics are assured [47]. Borophene has recently attracted a lot of attention due to its promising electronic properties. Borophene exhibits metallic properties [32,48], and ohmic contacts to conventional semiconductors can be realized, which leads to a significant improvement in the performance of electronic devices [49]. In the design of high-capacity and lightweight ion batteries, borophene is a promising anode material [50,51], and it is predicted that borophene with *β*_12_ phase and χ_3_ phase may have applications in Li- and Na-ion batteries [52].

Due to the unique physics of relativistic Dirac fermions, Dirac materials have attracted great attention [53,54]. The Dirac fermions in *β*_12_ borophene have been predicted in first-principles calculations [55,56] and observed by angle-resolved photoemission spectroscopy in the experimental studies [55]. One of the most important ultrarelativistic phenomena of Dirac fermions is Klein tunneling, which describes a perfect transmission of Dirac fermions at the normal incidence irrespective of the barrier height and width. Although Klein tunneling was proposed in 1929 [57], its experimental observation has only become possible since the discovery of monolayer graphene in 2004 [11]. In 2006, Katsnelson predicted Klein tunneling in graphene by designing an n-p-n junction, which has attracted a lot of experimental studies [58,59,60].

In the present work, we investigate theoretically the transport properties of Dirac electrons in *β*_12_ borophene based on the tight-binding model. Klein tunneling has been predicted in 8-Pmmn borophene in several studies [61,62], but work has rarely been done in *β*_12_ borophene due to the complexity of its tight-binding Hamiltonian. The non-equilibrium Green’s function (NEGF) was developed to study electron transport in bulk borophene, which is not limited to low-energy excitation. This method can be used in the studies of quantum transport in bulk graphene-like materials and provides accurate results without the low-energy limitation.

In Figure 1a, we show the crystal structure of *β*_12_ borophene. There are five atoms in a unit cell, as illustrated in Figure 1b, and the motion of electrons in borophene can be described by a five-band tight-binding Hamiltonian. Due to the perfect planar structure of *β*_12_ borophene, the electronic properties are well described by the tight-binding Hamiltonian in terms of p_z_ orbitals only [55]. The tight-binding Hamiltonian of *β*_12_ borophene with an external potential can be written as [55,63]
(1)$$H=\sum_{i}\varepsilon_{i}c_{i}^{+}c_{i}+\sum_{ij}^{n.n.}t_{ij}c_{i}^{+}c_{j}+U(x)$$
where $c_{i}^{+}$($c_{i}$) is the creation (annihilation) operator. The first term denotes the onsite energies for the five sites in a primitive cell, which is given by
(2)$$\begin{array}{l} \varepsilon_{a}=\varepsilon_{e}=0.196\;\mathrm{eV},\;\varepsilon_{b}=\varepsilon_{d}=-0.058\;\mathrm{eV},\\ \varepsilon_{c}=-0.845\;\mathrm{eV}, \end{array}$$

The second term denotes the nearest neighbor hopping in the lattice of *β*_12_ borophene, and the hopping energies are summarized as
(3)$$\begin{array}{l} t_{ab}=t_{de}=-2.04\;\mathrm{eV},\;t_{ac}=t_{ce}=-1.79\;\mathrm{eV},\\ t_{ae}=-2.12\;\mathrm{eV},\\ t_{bc}=t_{cd}=-1.84\;\mathrm{eV},t_{bd}=-1.91\;\mathrm{eV}, \end{array}$$

In this study, we investigate the transport properties of bulk *β*_12_ borophene, so the Bloch theorem is imposed along the transverse direction (see Figure 1c). Due to the periodic boundary conditions, the hopping between atoms a and e is modified to $t_{ae}^{*}=e^{ik_{y}W}$, where $e^{ik_{y}W}$ is the Bloch phase factor. In the present theoretical model shown in Figure 1c, the periodicity is W = 3a. In Figure 1d, we show the definition of the incident angle $\theta =\arctan(k_{y}/k_{x})$. When electrons incident into the scattering region (region II), different scattering processes such as Klein tunneling can be found, which can be revealed by the transmission probability of electrons. The transmission probability of electrons can be given by the non-equilibrium Green’s function (NEGF). The non-equilibrium Green’s function has a great advantage in dealing with the transmission probability of electrons passing through the potential barrier. By adopting the NEGF framework, we seamlessly integrate electron–electron interactions and non-equilibrium conditions, thereby achieving a comprehensive understanding of the transport phenomena within our system. Moreover, the NEGF approach empowers us to compute the retarded Green’s function, an indispensable tool for determining transmission probabilities throughout the system. Employing periodic boundary conditions for the scattering region Hamiltonian, alongside incorporating self-energy contributions from the leads, ensures the precise modeling of the system’s transport properties. So, the transmission probability of electrons can be given:(4)$$T(E,k_{y})=Tr\left\{\Gamma_{R}(E,k_{y})G^{r}(E,k_{y})\Gamma_{L}(E,k_{y})G^{a}(E,k_{y})\right\}$$
where $\Gamma_{L(R)}$ is the level-broadening matrix, which is calculated from the lead self-energies as $\Gamma_{L(R)}=i[\Sigma_{L(R)}^{r}-\Sigma_{L(R)}^{a}]$, and $G^{r(a)}$ is the retarded (advanced) Green’s function, which can be obtained by
(5)$$G^{r(a)}={\left[EI-H-\Sigma_{L}^{r(a)}-\Sigma_{R}^{r(a)}\right]}^{-1}$$
where *H* is the Hamiltonian with periodic boundary conditions for the scattering region, and $\Sigma_{L}^{r(a)}(\Sigma_{R}^{r(a)})$ is the self-energy due to the interaction with the left (right) lead. The conductance at zero temperature can be obtained from the Landauer–Buttiker formalism as [64,65]
(6)$$G_{P/AP}(E_{F})=\frac{4e^{2}}{h}\int_{-E_{F}}^{E_{F}}T_{P/AP}(E_{F},k_{y})\frac{dk_{y}}{2\pi /L_{y}}$$
where $G_{0}\;{=2e}^{2}L_{y}/\pi h$ with $L_{y}$ are the transverse length of the sample.

## 2. Results and Discussion

The band structure of bulk borophene for $k_{y}=0$ is shown in Figure 2a. In a unit cell, we have five boron atoms, so we notice that there are five bands in the band structure for the periodical structure with $k_{y}=0$. In this study, we mainly focus on the quantum transport in low energy, so we zoom in on the low energy range (see the red rectangular section).

The enlargement of the red rectangular part of Figure 2a is shown in Figure 2b. We can see that the band structure is similar to graphene, in which two conically shaped valleys are formed by the valence and conduction bands. The valley degree of freedom in borophene may also be used to carry information for valleytronics [66,67], which is saved for future studies. There are two gapless Dirac points, K and K’, in the Brillouin zone; however, we note that the Dirac points are 0.1937eV above the Femi energy E_F_, which is different from graphene. Correspondingly, the relevant parameters of the system are defined and marked in Figure 2c, e.g., the energy E is the value above the Dirac points. The definition of the momentum $k_{x}$ is presented in Figure 2d. For a given energy E and $k_{y}$, we can calculate $k_{x}$ from the periodic Hamiltonian, and the incident angle of electrons can be obtained $\theta =\arctan(k_{y}/k_{x})$.

In Figure 3, we show the polar plots of the transmission probability for a single potential barrier. We fix the energy at E = 80 meV and change the ratio of E/V to check the transmission properties, where four different widths of the potential barrier, D = 80 nm, 110 nm, 140 nm, and 170 nm are considered. It is clear that the potential barrier becomes totally transparent (T(θ=0)=1) for normally incident electrons. Even when the barrier heights and widths are changed, normally incident electrons can still propagate with 100% efficiency. This unique feature is a manifestation of the Dirac fermions, which is related to Klein tunneling. Klein tunneling is a phenomenon that cannot be observed in nonrelativistic electrons, and it has been explored and explained in graphene. It is obvious that such a phenomenon driven by the Dirac fermions can also be observed in *β*_12_ borophene.

In Figure 3a, we present the transmission probability for three E/V ratios with the barrier width D fixed to be 80 nm. For the large E/V ratio, we only have nonzero transmission in a small angle region. This feature can also be observed in Figure 3b–d, where a larger width of the barrier is considered. Such a feature can be used to control the guiding of electrons by the gate, which may find applications in electronic fiber or electron waveguides. We also note that the barrier can be perfectly transparent (T(θ)=1) at some other incident angles, for example, the transmission is 100% at θ=30° and 62° in Figure 3a. These transmission peaks are caused by the Fabry–Perot resonances in the barrier, which occur due to the interference of electron waves that are reflected back and forth inside the barrier. Since the transmission peaks are caused by the interference of the electron waves, the number of the transmission peaks depends on the barrier width strongly. Comparing the above four pictures, we can see that the number of transmission peaks increases as the width of the barrier increases.

We proceed with the investigation of a single potential barrier and a double potential barrier, aiming to compare their distinctions. Figure 4 illustrates the transmission probability’s dependence on the Fermi energy and the incident angle. To examine the transmission properties, we maintain the barrier height at V = 200 meV while varying the Fermi energy and the incident angle. In Figure 4a–c, we explore three distinct widths of the single potential barrier, namely, D = 30 nm, 40 nm, and 50 nm, respectively. Additionally, we create two potential barriers using a barrier width of 40 nm and a barrier height of V = 200 meV, varying the width of separation between the two barriers (L = 40 nm, 80 nm, and 120 nm) in Figure 4d–f. Notably, both the single and double potential barriers exhibit total transparency (T(θ=0)=1) when electrons are incident normally, indicating the presence of Klein tunneling in both cases.

Examining the results further, we observe transmission resonances (Fabry–Perot resonances) at specific Fermi energies for a fixed incident angle, and these resonant positions shift when altering the incident angle, as shown in Figure 4a. The same phenomenon is observable in Figure 4b,c, where larger barrier widths are considered. It is also worth noting that the number of resonances increases with an increase in barrier thickness, as evident in Figure 4a–c. Furthermore, in Figure 4d–f, apart from the barrier resonances, additional resonances arise due to wave reflections between barriers. These can be distinguished from the former resonances by comparing them with Figure 4b, which corresponds to a single barrier with a width of 110 nm. Notably, a pronounced dependence of transmission probability on the incident angle is observed around 270~300 meV, with peaks separated by regions of zero transmission occurring at large incident angles. The number of resonances increases as the distance between two potential barriers widens in Figure 4d–f.

Despite the distinct resonant behavior in the single and double potential barriers, both cases result from the interference of electron waves. Given the similarity in energy band structures between graphene and β_12_ boronene, these resonances may also be observed in graphene.

In Figure 5a, we present the transmission probability as a function of the Fermi energy and the incident angle. By maintaining the barrier height at V = 200 meV and the barrier width at 158 nm, we explore the transmission properties across varying incidence angles and energies. Notably, the transmission resonances’ branching behavior becomes inwardly concave with an increase in the Fermi energy. Moreover, when the incident electron energy is low, we observe an intriguing phenomenon wherein the axis of the resonant branch, parallel to the angle of incidence, can be represented by the white line in Figure 5a. This specific situation is further illustrated in Figure 5b.

In Figure 5b, we notice a nearly perfect transmission (T(θ)=1) for all incidence angles, resembling the concept of “super” transmission. This high transmission efficiency remains significant for all angles except those close to 90 degrees, where the perfect transmission (T(θ)=1) slightly diminishes. This unique characteristic is referred to as “super-Klein tunneling”, a phenomenon observed in both the Lieb and Dice lattice models as described in publication [64]. It is worth noting that the super-Klein tunneling in β_12_ borophene occurs at incident energies near the Dirac points, adding to its distinctive properties.

In Figure 6, we present the conductance G/G_0_ as a function of the width of the potential barriers D, considering various ratios of E/V. The energy is kept fixed at E = 80 meV, and we modify the E/V ratio to investigate the conductance properties. The E/V ratio signifies the relationship between the incident electron energy and the barrier height. Notably, the conductance shows a clear decrease as the width of the potential barriers D increases. This behavior can be attributed to a reduction in the total number of electrons passing through the barriers as the barrier widths widen, resulting in a lower conductance.

For cases with small E/V ratios, a noticeable oscillatory behavior is observed in the conductance. These oscillations are regular within a specific range of small barrier widths but become irregular in the range of large barrier widths. This oscillatory pattern is related to Fabry–Perot resonances, where the incident electrons undergo multiple reflections between the barriers, leading to constructive and destructive interference effects on the transmission. On the other hand, when the E/V ratio is large, no oscillations are present in the conductance. This is due to the fact that only a small angular region exhibits nonzero transmission, which results in the absence of oscillations in the conductance.

Overall, the conductance behavior in Figure 6 is intricately linked to the width of the potential barriers, the E/V ratio, and the occurrence of Fabry–Perot resonances, providing valuable insights into the electron transport properties within the system.

## 3. Conclusions

In our investigation of β_12_ borophene, we have thoroughly explored the transmission properties across both a single potential barrier and two potential barriers. Remarkably, when electrons are incident perpendicular to the interface, we have observed that the potential barriers exhibit total transparency, regardless of variations in their heights and widths. Interestingly, at specific values of incident energy and barrier width, we have noticed a fascinating phenomenon wherein the transmission rate becomes independent of the incident angle. This characteristic closely resembles the concept of “super transmission” previously reported for the dice lattice.

Moreover, in scenarios with large E/V ratios, we have identified nonzero transmission occurring within a small angle region. This intriguing feature could have practical applications in electronic fiber or electron waveguide systems. For cases with small E/V ratios, we have observed the presence of Fabry–Perot resonances in β_12_ borophene, akin to what is seen in graphene and the dice lattice. Notably, by examining the resonant behavior in both the single and double potential barrier configurations, we have been able to distinguish between resonance phenomena associated with a single barrier and those arising from interactions between two barriers.

These discoveries significantly contribute to our understanding of electron transmission properties in β_12_ borophene and offer valuable insights into potential applications in nanoelectronics and waveguide technologies.

## Figures and Tables

**Figure 1 nanomaterials-14-00790-f001:**
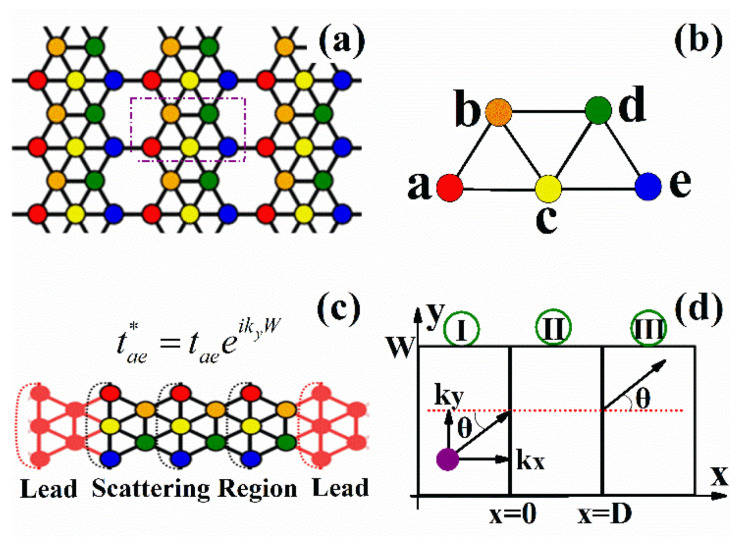
(**a**) The lattice structure of *β*_12_ borophene. The unit cell is indicated by a purple dashed rectangle. (**b**) The unit cell contains five atoms, which are labeled as a, b, c, d, and e. (**c**) A minimum tight-binding model with periodic boundary conditions for bulk borophene. The hopping between atoms a and e is modified and indicated as $t_{ae}^{*}$, where $e^{ik_{y}W}$ is the Bloch factor with W = 3a. (**d**) Schematic of the scattering of electrons by a potential barrier and the definition of the incident angle.

**Figure 2 nanomaterials-14-00790-f002:**
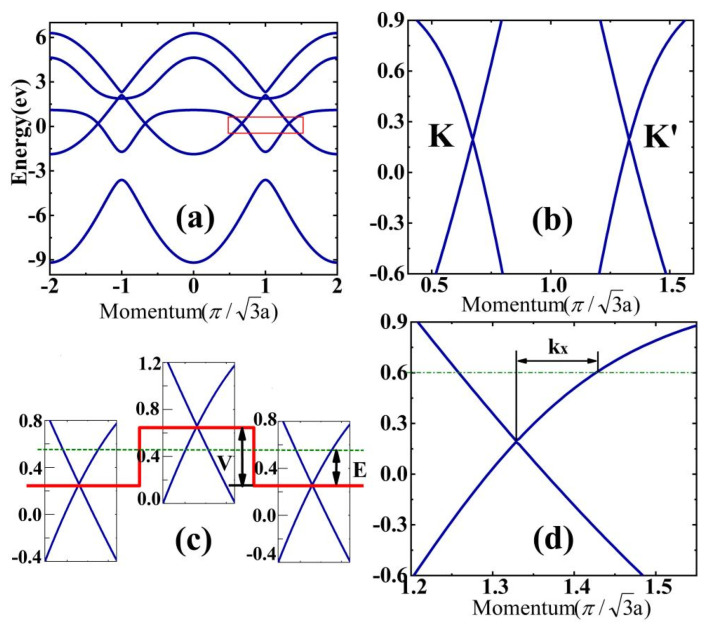
(**a**) The band structures of bulk borophene with $k_{y}=0$. (**b**) An enlarged schematic of the red rectangular part of (**a**); two gapless Dirac points, K and K’, are indicated. (**c**) Schematic illustrations of the n-p-n junction. The definition of the incident energy E and the height of the potential barrier V are marked. (**d**) The definition of the momentum $k_{x}$.

**Figure 3 nanomaterials-14-00790-f003:**
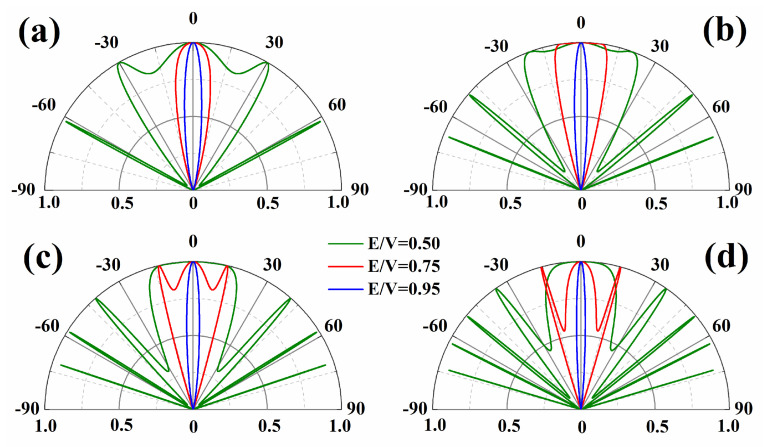
Polar plots of the transmission probability for a potential barrier. The widths of the potential barriers in panels (**a**–**d**) are D = 80 nm, 110 nm, 140 nm, and 170 nm, respectively. The energy is chosen to be E = 80 meV, and three cases with E/V = 0.5 (green), 0.75 (red), and 0.95 (blue) are considered.

**Figure 4 nanomaterials-14-00790-f004:**
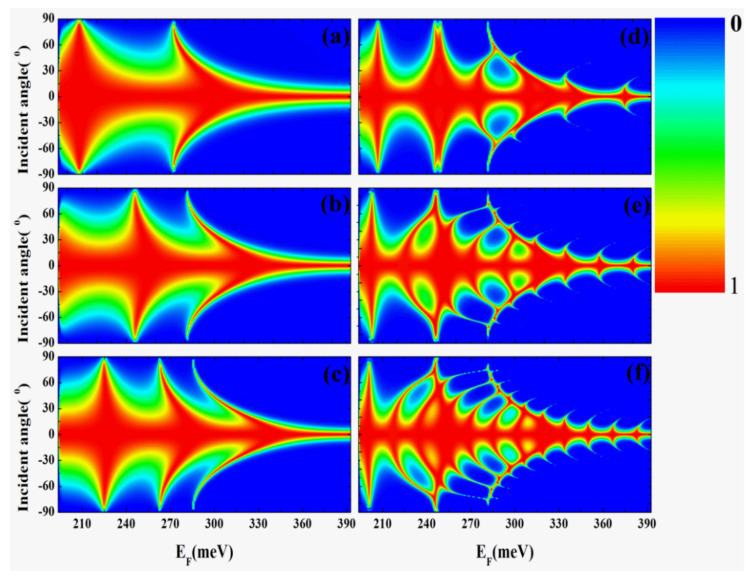
Contour plots of the transmission probability as a function of incidence angle and energy. Panels (**a**–**c**) are the transmission of borophene with a single potential barrier with the barrier height of V = 200 meV, and the barrier width from top to bottom: D = 30 nm, 40 nm, 50 nm. Panels (**d**–**f**) are the transmission of borophene with double potential barriers, and the height and width of barriers are 200 meV and D = 40 nm, respectively. The separation between the two barriers from top to bottom: L = 40 nm, 80 nm, and 120 nm.

**Figure 5 nanomaterials-14-00790-f005:**
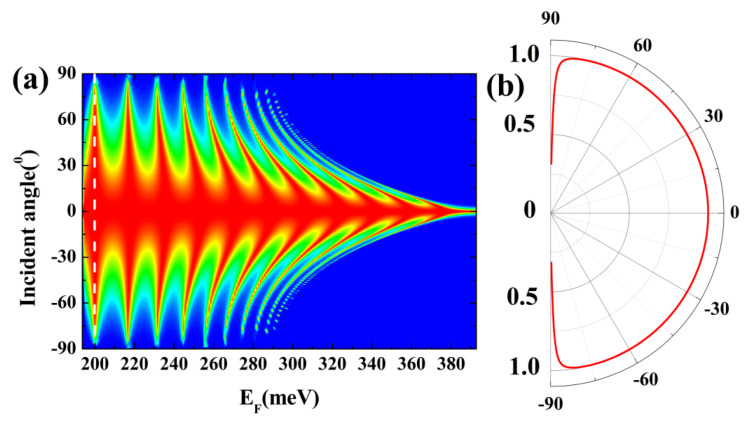
(**a**) Contour plots of the transmission probability as a function of incidence angle and energy. The height and the width of the potential barriers are 200 meV and D = 158 nm, respectively. (**b**) When the Fermi energy is 200 meV, the transmission probability depends on the angle of incidence as indicated by the white dashed line in (**a**).

**Figure 6 nanomaterials-14-00790-f006:**
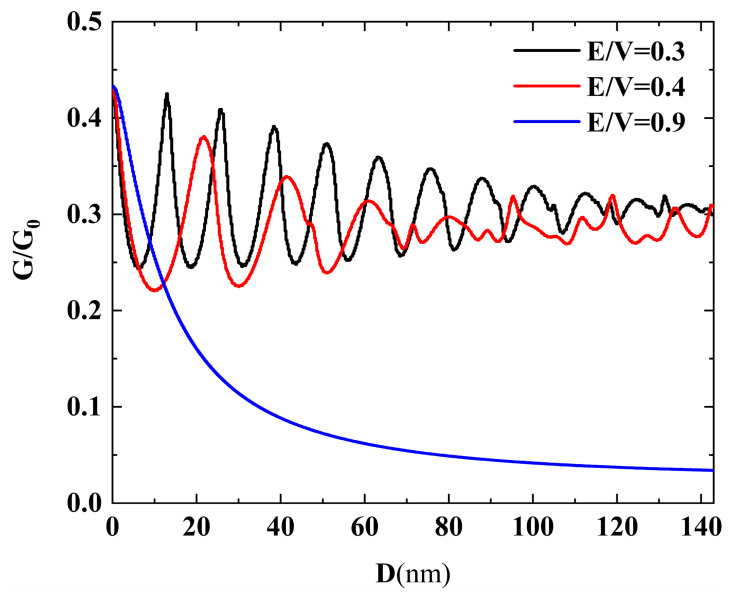
The conductance G/G_0_ as a function of the width of the potential barriers D. The energy is chosen to be E = 80 meV, and three cases, with E/V = 0.3 (black), E/V = 0.4 (red), and E/V = 0.9 (blue) are considered.

## Data Availability

Data are contained within the article.

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
