# Peer review of "Klein Tunneling in *β*_12_ Borophene"

_nanomaterials, 2024, doi:10.3390/nano14090790_

Round 1

Reviewer 1 Report

Comments and Suggestions for Authors

This manuscript reports on observation of the Klein tunneling in β12 borophene. Studying the transport properties of bulk β12 borophene is a task within an important topic which is not well investigated while the methodology used in the present work is quite adequate and adapted to the complexity of phenomena involved. Results obtained are interesting for the community and may be useful for the field of borophene as well as of other B-based  nanostructured materials. This work is also interesting methodologically because of the specificities of electronic structure of β12 borophene are discussed on a comparative scale and explained well.

The work raises only some minor textual concerns related to insufficient contextualization of the present results. These points amount to a minor revision before the acceptance of manuscript for publication:

1: Abstract should briefly emphasize the tight-binding model and the complexity of the tight-binding Hamiltonian involved.

2: The non-equilibrium Green's function (NEGF)involvement should be better contextualized since that’s another advantage of the concept as employed in the present work.

3: Dynamic stability of β12 borophene should be implicitly mentioned as an inherent prerequisite for all subsequent electronic structure and transport studies and analises.

4: The authors fail to mention the existing literature dedicated to novel inherently nanostructured and complex systems (including similar to those studied in the present work), whereby electronic and transport properties have ALSO been successfully addressed by ab initio methods, namely CrystEngComm 23 (2021) 6661-6667; The Journal of Physical Chemistry C 118 (2014) 11377-11384. Such existing literature should be reflected in the introduction.

Comments on the Quality of English Language

The manuscript still needs a comprehensible stylistic and grammatical revision.

Reviewer 2 Report

Comments and Suggestions for Authors

The manuscript reports the theoretical study of Klein tunneling in beta12 borophene.
The authors described the electronic states in the system within the tight-binding model
and calculated the transmission probabilities and conductance through potential barriers
using the non-equilibrium Green's function method.
Beta12 borophene has Dirac corns near the Fermi level, and thus the charge carriers
should exhibit Klein tunneling through potential barriers formed in beta12 borophene.
In this sense, the calculated results are easily expected from the band structure,
but I do not see the significance of revisiting the Klein tunneling in 2d material,
which has been thoroughly investigated so far in the context of Dirac fermions in graphene.
For this reason, I do not think that the present manuscript deserves publication
unless qualitative differences from the Klein tunneling e.g. in graphene are discussed.

Comments on the Quality of English Language

The English text is well except for a few minor typos.

Reviewer 3 Report

Comments and Suggestions for Authors

The authors performed a theoretical study of the properties of β12 borophene. This two-dimensional material is of practical interest and is currently being actively studied. Thus, the work is relevant. There are a few comments. However, the following technical issues need to be addressed prior to publication:

1. The authors calculated the band structure of β12 borophene. The authors write that it is similar to graphene. Currently, there are calculations for β12 borophene within the framework of first-principles calculations. Did the authors compare their results with other results?

2. The authors obtained theoretical results using the model. Is it possible to make a direct comparison with experimental data? Are there similar experimental data? How to make sure that the results obtained are correct. How to make sure that the model and approximations are correct?

Round 2

Reviewer 3 Report

Comments and Suggestions for Authors

Accept